# Bicarbonate-Dependent Detoxification by Mitigating Ammonium-Induced Hypoxic Stress in *Triticum aestivum* Root

**DOI:** 10.3390/biology13020101

**Published:** 2024-02-05

**Authors:** Xiao Liu, Yunxiu Zhang, Chengming Tang, Huawei Li, Haiyong Xia, Shoujin Fan, Lingan Kong

**Affiliations:** 1Crop Research Institute, Shandong Academy of Agricultural Sciences, Jinan 250100, China; 2021020839@stu.sdnu.edu.cn (X.L.); 13116012750@163.com (Y.Z.); 2022020865@stu.sdnu.edu.cn (C.T.); lily984411@126.com (H.L.); haiyongxia@cau.edu.cn (H.X.); 2College of Life Science, Shandong Normal University, Jinan 250014, China

**Keywords:** dioxygenase, fermentation, root, TCA cycle, wheat

## Abstract

**Simple Summary:**

Ammonium (NH_4_^+^) is usually toxic to plant growth when used as the sole or dominant N source. Exploring the underlying molecular mechanisms of NH_4_^+^ toxicity and how to minimize NH_4_^+^ toxicity may greatly benefit crop productivity. In this study, the underlying mechanism of NH_4_^+^ toxicity and bicarbonate (HCO_3_^−^)-dependent alleviation in wheat was investigated. Comprehensive transcriptomic and physiological analyses suggested that NH_4_^+^ nutrition alone stimulated fermentation and glycolysis, promoted the activity of alternative respiratory pathways, suppressed TCA cycle pathways, and reduced ATP synthesis; adding HCO_3_^−^ relieved the toxic effects of NH_4_^+^ nutrition. Our results reveal the importance of C and N interactions for alleviating NH_4_^+^ toxicity, likely by mitigating root hypoxic stress. As the first report on the hypoxic stress triggered by NH_4_^+^ treatment, this study provides novel insights into the mechanisms of NH_4_^+^ toxicity and its alleviation, which may present potential solutions for improving the nitrogen use efficiency in wheat.

**Abstract:**

Ammonium (NH_4_^+^) toxicity is ubiquitous in plants. To investigate the underlying mechanisms of this toxicity and bicarbonate (HCO_3_^−^)-dependent alleviation, wheat plants were hydroponically cultivated in half-strength Hoagland nutrient solution containing 7.5 mM NO_3_^−^ (CK), 7.5 mM NH_4_^+^ (SA), or 7.5 mM NH_4_^+^ + 3 mM HCO_3_^−^ (AC). Transcriptomic analysis revealed that compared to CK, SA treatment at 48 h significantly upregulated the expression of genes encoding fermentation enzymes (pyruvate decarboxylase (PDC), alcohol dehydrogenase (ADH), and lactate dehydrogenase (LDH)) and oxygen consumption enzymes (respiratory burst oxidase homologs, dioxygenases, and alternative oxidases), downregulated the expression of genes encoding oxygen transporters (PIP-type aquaporins, non-symbiotic hemoglobins), and those involved in energy metabolism, including tricarboxylic acid (TCA) cycle enzymes and ATP synthases, but upregulated the glycolytic enzymes in the roots and downregulated the expression of genes involved in the cell cycle and elongation. The physiological assay showed that SA treatment significantly increased PDC, ADH, and LDH activity by 36.69%, 43.66%, and 61.60%, respectively; root ethanol concentration by 62.95%; and lactate efflux by 23.20%, and significantly decreased the concentrations of pyruvate and most TCA cycle intermediates, the complex V activity, ATP content, and ATP/ADP ratio. As a consequence, SA significantly inhibited root growth. AC treatment reversed the changes caused by SA and alleviated the inhibition of root growth. In conclusion, NH_4_^+^ treatment alone may cause hypoxic stress in the roots, inhibit energy generation, suppress cell division and elongation, and ultimately inhibit root growth, and adding HCO_3_^−^ remarkably alleviates the NH_4_^+^-induced inhibitory effects on root growth largely by attenuating the hypoxic stress.

## 1. Introduction

Nitrogen (N) is one of most essential nutrients for plant growth and development and is involved in the biosynthesis of proteins, nucleic acids, chlorophyll, and several hormones [1,2,3]. Therefore, appropriate N fertilizer use is critical in increasing crop yield and improving the quality of agricultural products. Ammonium (NH_4_^+^) and nitrate (NO_3_^−^) are the two main N sources for plants [4,5,6]. NH_4_^+^ can be more directly assimilated by plant cells, while NO_3_^−^ uptake and reduction (i.e., conversion of NO_3_^−^ to NH_4_^+^ via the actions of nitrate reductase and nitrite reductase) consume large amounts of ATP and reducing equivalents, and the resulting NH_4_^+^ is then used by plants [5,7,8]. Therefore, it is widely acknowledged that NH_4_^+^ is the preferred N source with respect to energy cost.

To increase crop yields, farmers tend to apply excessive N fertilizers; however, only 30–40% of the fertilizer is estimated to be taken up by plant roots [8,9]. Moreover, excess N fertilizer application often suppresses crop growth, decreases kernel yield, and causes environmental pollution [10]. In particular, when using NH_4_^+^ as the sole N source, most plants exhibit severe growth retardation, leaf chlorosis, a lower net photosynthetic rate, and other toxic symptoms, commonly referred to as NH_4_^+^ syndrome [11,12]. Therefore, optimizing N fertilization can improve nitrogen use efficiency (NUE), reduce the cost of N inputs, and decrease environmental pollution [13].

An increasing amount of research has been conducted with the aim of unraveling the mechanisms underlying NH_4_^+^ syndrome in plants, and numerous studies have proposed hypotheses to explain NH_4_^+^ toxicity, including ion imbalance of the essential cations (K^+^, Ca^2+^, Mg^2+^, Fe^2+/3+^, Zn^2+^, SO_4_^2−^, and PO_4_^3−^) [6,14,15], acidification of the apoplast and rhizosphere and transient alkalization of the cytosol [16,17], ATP overconsumption and waste [14,18], carbon (C) reserve deprivation caused by the excess consumption of soluble sugars during NH_4_^+^ assimilation [19], the accumulation of amino acids and depletion of organic acids [20], imbalanced hormone interactions [6,21], and reduced photosynthesis [14]. However, several hypotheses are controversial and the mechanisms of NH_4_^+^ phytotoxicity are not fully understood [15,22].

Many studies have been conducted to elucidate the underlying mechanisms by which exogenous substances alleviate NH_4_^+^ toxicity. It was found that elevated potassium (K^+^) reduced futile NH_4_^+^ cycling on the plasma membrane in rice (*Oryza sativa* L.) [23], and decreased vacuolar H^+^-ATPase activity and inhibited NH_4_^+^ accumulation in *Arabidopsis thaliana* roots [24]. Low levels of NO_3_^−^ attenuate NH_4_^+^ toxicity by upregulating *ACLA-3* (encoding ATP-citrate lyase A-3) and increasing the production of several key metabolites in the tricarboxylic acid (TCA) cycle in *Panax notoginseng* [25] and wheat [7]. SnRK1.1 allows SLAH3-mediated NO_3_^−^ efflux by phosphorylating the C-terminal of SLAH3 at site S601, thereby alleviating high-NH_4_^+^/low-pH stress in *Arabidopsis thaliana* [26]. Exogenous *α*-ketoglutarate (KGA), a key C skeleton for N assimilation, alleviates NH_4_^+^ stress in tomato (*Solanum lycopersicum* L.) [27]. Silicon (Si) alleviates NH_4_^+^ toxicity by accelerating NH_4_^+^ assimilation via the actions of glutamine synthetase, glutamate synthase, and glutamate dehydrogenase in cabbage (*Brassica campestris* L.) [28] or increasing the shoot cytokinin content in tomato [29].

When used as the dominant N source, NH_4_^+^ stimulates respiratory O_2_ consumption in *Arabidopsis*, barley (*Hordeum vulgare* L.), wheat, and maize (*Zea mays* L.) to meet the needs of ATP, resulting in higher carbon dioxide (CO_2_) evolution [30,31,32]. Therefore, the reduced NH_4_^+^ toxicity after the exogenous addition of CO_3_^2−^ [33], HCO_3_^−^ [34], or CO_2_ [35] may be explained by improved carbohydrate accumulation, balanced C and N metabolism, and a greater ability to cope with the depletion of organic acids [25]. Notably, these changes are involved in the TCA cycle, where intermediates, mainly KGA, can be used as C skeletons for NH_4_^+^ assimilation [36,37].

Wheat is one of the most important food crops, and about one third of the global population currently consumes wheat [38]. Improving NUE is important to improve grain yield and processing quality and reduce the cost of wheat production [39]. In this study, comprehensive transcriptomic and physiological analyses were conducted to investigate the underlying mechanisms of NH_4_^+^ toxicity and its HCO_3_^−^-dependent alleviation in wheat. Our results show that NH_4_^+^ nutrition alone stimulated fermentation and glycolysis, promoted the activity of alternative respiratory pathways, suppressed TCA cycle pathways, and reduced ATP synthesis; adding HCO_3_^−^ relieved the toxic effects of NH_4_^+^ nutrition. Our results reveal the importance of C and N interactions for alleviating NH_4_^+^ toxicity, likely by mitigating root hypoxic stress.

## 2. Materials and Methods

### 2.1. Plant Material and Growth Conditions

Seeds of wheat (cultivar Jimai 22) were surface-sterilized using 70% ethanol for 45 s and washed 5 times with distilled water. The sterilized seeds were then germinated on moist filter paper placed inside Petri dishes at 23 °C. After 3 days, the uniform-sized seedlings were transferred to black plastic pots with dimensions of 10 cm × 8 cm × 5 cm (length, width, height) containing distilled water and grown in a growth chamber at 25 °C/21 °C (day/night) under 14 h/10 h (light/dark) for 5 days under the following conditions: light intensity 450 µmol m^−2^ s^−1^ and humidity 70 ± 5%. Each pot contained 15 plants and the distilled water was renewed every 2 days.

In our preliminary experiment, we found that the roots of 8-day-old seedlings grew best in half-strength Hoagland nutrient solution containing 7.5 mM NO_3_^−^ (Appendix A), and 7.5 mM NH_4_^+^ showed significant inhibition of root growth compared with 7.5 mM NO_3_^−^. An experiment using gradient concentrations showed that 3 mM HCO_3_^−^ significantly improved the root growth of wheat seedlings fed 7.5 mM NH_4_^+^. Therefore, the following experiment was conducted using 8-day-old seedlings that were fixed on polystyrene plates and hydroponically cultured in half-strength Hoagland nutrient solution containing 7.5 mM NO_3_^−^ (CK, applied as KNO_3_ and Ca(NO_3_)_2_), 7.5 mM NH_4_^+^ (sole ammonium (SA), applied as 7.5 mM NH_4_Cl), or 7.5 mM NH_4_^+^ + 3 mM HCO_3_^−^ (ammonium and bicarbonate (AC), applied as 7.5 mM NH_4_Cl and 3 mM KHCO_3_). The solutions were renewed every 2 days. Each treatment was repeated in triplicate. Potassium in the nutrient solution was balanced by the addition of K_2_SO_4_.

### 2.2. Measurement of Plant Fresh Weight (FW)

Wheat seedlings were collected at 24, 48, 72, and 96 h after treatment and separated into shoots and roots, which were dried with absorbent paper and immediately weighed. Three biologically independent experiments, each with three replicates, were conducted to calculate the net increase in FW.

### 2.3. Transcriptome Sequencing

#### 2.3.1. RNA Extraction and Detection

Fresh roots (approximately 0.1 g) were fully ground in a mortar with 1 mL RLT and 100 µL PLANTaid at room temperature. The homogenate was transferred to a centrifuge tube, vigorously shaken for 15 s, and centrifuged at 13,000 rpm for 5 min. Then, 450 µL of supernatant was transferred to a new centrifuge tube, and the volume of half absolute ethanol was added, blown, and mixed. The mixture was added to an adsorption column for RA (the adsorption column was placed in the collection tube) and centrifuged at 13,000 rpm for 60 s, and the waste liquid was abandoned. Total RNA extracted from roots underwent RNA-Seq analysis at Novogene (Beijing, China). Three biological replicates of each sample were used for RNA-Seq analysis. RNA quality and quantity were checked using a spectrophotometer (NanoDrop ND-1000 UV-Vis spectrophotometer; Nanodrop Technologies, Wilmington, DE, USA). The integrity of the final RNA samples was checked by denaturing gel electrophoresis on 1.4% (*w*/*v*) formaldehyde agarose gels, and the concentration was determined photometrically (NanoDrop). Purified RNA was treated with a Turbo DNase-free kit. cRNA synthesis and labeling, array hybridization, and scanning were performed at imaGenes GmbH (Berlin, Germany).

#### 2.3.2. Library Construction and Quality Inspection

There are 2 ways to construct a library: ordinary NEB construction and chain-specific construction. The NEBNext^®^ Ultra™ RNA Library Prep Kit from Illumina^®^ was used to build the library. After RNA library construction, initial quantification was characterized on 1% agarose gels and examined using the NanoPhotometer^®^ spectrophotometer (Implen, Westlake Village, CA, USA). RNA concentrations were measured using a Qubit^®^ RNA Assay Kit in a Qubit^®^ 2.0 Fluorimeter (Life Technologies, Carlsbad, CA, USA). The RNA integrity number was analyzed by accurate detection of RNA integrity and library insert size using an Agilent Bioanalyzer 2100 system (Agilent Technologies, Santa Clara, CA, USA). After the insert size was determined, effective library concentrations were accurately quantified by qRT-PCR (above 2 nM) to ensure quality.

#### 2.3.3. Sequencing

After inspection, libraries were pooled and sequenced against the effective concentration and target data volume. Four fluorescently labeled dNTP, DNA polymerase, and adapter primers were amplified to the sequenced flow cells. When the sequencing cluster extended the complementary strand, each fluorescently labeled dNTP was released. The sequencer captured the fluorescence signal and the light signal was converted to the sequencing peak through the computer software to obtain the sequence information of the fragments to be measured.

#### 2.3.4. Data Quality Control

The image data of sequencing fragments measured using a high-throughput sequencer were converted into sequence data (reads) by CASAVA bases, which mainly contained the sequence information of the sequencing fragments and the corresponding quality information. The sequencing error rate distribution was checked and Q20, Q30, and GC contents were determined, and a small number of reads with low sequencing quality were filtered out of the raw data to obtain clean reads with high quality for subsequent analysis (Appendix A). The clean reads were quickly and accurately aligned to the reference genome using HISAT2-2.1.0 software to obtain the mapping information of reads on the reference genome.

### 2.4. Analysis of Total Differentially Expressed Genes (DEGs)

After quantifying gene expression, we performed a statistical analysis of their expression data to screen the samples for genes with significantly different expression levels in different states. The original read count was first standardized (normalized) and mainly corrected for sequencing depth. The statistical model then calculated the hypothesis testing probability (*P*_adj_) and performed multiple-hypothesis test correction to obtain the false discovery rate (FDR, commonly notated as *P*_adj_). Finally, the number of DEGs for each comparative combination was counted and screened to analyze the expression of target genes.

### 2.5. GO and KEGG Enrichment Analysis of DEGs

Gene Ontology (GO) proteome annotation and the Kyoto Encyclopedia of Genes and Genomes (KEGG) database were used to annotate pathways, and they were derived from the online DAVID Bioinformatics tools (https://david.ncifcrf.gov/home.jsp; accessed on 28 March 2023). First, identified gene IDs were converted to Entrez gene IDs and then mapped to GO IDs by gene ID. GO is a comprehensive database describing gene function, divided into three parts: biological process (BP), cellular component (CC), and molecular function (MF). Correction for multiple hypothesis tests was carried out by using standard false discovery rate control methods. In the following discussion, GO functional enrichment was determined with *P*_adj_ < 0.05 as the threshold for significance. KEGG is a comprehensive database that integrates information on genomic, chemical, and systematic functions. KEGG pathway enrichment analysis of DEGs was performed with *P*_adj_ < 0.05 as the threshold for significance.

### 2.6. Enzymatic Assays

Fresh root tissue (approximately 1.0 g) was added with 1.6 mL of pre-cooled phosphate buffer (1 mM AsA, 3 mM β-mercaptoethanol, 0.5 mM PMSF, 2% PVP, 1 mM EDTA, pH 7.8). The mixture was ground with liquid nitrogen, the extract was centrifuged at 4 °C at 12,000× *g* for 20 min, and the supernatant was used for the determination of enzyme activity. Complex V, pyruvate decarboxylase (PDC), alcohol dehydrogenase (ADH), lactate dehydrogenase (LDH), pyruvate kinase (PK), and pyruvate dehydrogenase complex (PDHC) activity was determined using commercial chemical detection kits (Comin, Suzhou Comin Biotechnology Co., Ltd., Suzhou, China) according to the instructions provided by the manufacturer. Absorbance measurement was performed using a 96-well microplate reader (Rayto RT-6100, Rayto Company, Shenzhen, China), and the corresponding calculation formula was used to calculate enzyme activity.

### 2.7. Ethanol and Organic Acid Determination

Fresh roots (approximately 2.0 g) were homogenized using a mortar and pestle with 10 mL of pre-chilled extracting solution (80% methanol *v*/*v*, 100 mM imidazole, pH 7.0) and then heated at 80 °C for 15 min. The homogenate was transferred into 2 mL Eppendorf tubes and centrifuged at 8000× *g* at 4 °C for 10 min, and then the supernatant was collected. The concentrations of ethanol, lactate (LA), pyruvate (Pyr), acetyl-CoA, citrate (CA), KGA, succinate, fumarate, malate, oxaloacetic acid (OAA), alanine (Ala), γ-aminobutyric acid (GABA), formate, tryptophan (Trp), tyrosine (Tyr), and phenylalanine (Phe) were determined using commercial chemical detection kits (Comin, Suzhou Comin Biotechnology Co., Ltd., Suzhou, China). The spectrophotometric value of the solution was measured using a spectrophotometer (Varian Cary 100) or a microplate reader (Rayto RT-6100, Rayto Company, Shenzhen, China). The metabolite concentrations were calculated according to the manufacturer’s instructions.

### 2.8. Measurement of ATP and ADP Content

Fresh roots (approximately 1.0 g) were homogenized using a mortar and pestle with 5 mL of extracting solution (96% ethanol, 0.1 M EDTA, pH 7) at 78 °C. This homogenate was heated in a boiling water bath for 1 min and filled with nitrogen for 10 min, and then the supernatant was diluted with available Tris-EDTA. The ATP and ADP content was measured on ice by chemiluminescent analysis using commercial chemical detection kits (Comin, Suzhou Comin Biotechnology Co., Ltd., Suzhou, China) according to the manufacturer’s instructions.

### 2.9. Quantitative Reverse Transcription Polymerase Chain Reaction (qRT-PCR)

qRT-PCR was conducted using TaqPro Universal SYBR qPCR Mastermix (Q712-02, Vazyme, Nanjing, China) according to the manufacturer’s instructions. Three independent biological repetitions were performed. qRT-PCR was performed under the following conditions: 95 °C for 3 min, followed by 40 cycles of 95 °C for 10 s, 58 °C for 30 s, and 72 °C for 30 s. The glyceraldehyde 3-phosphate dehydrogenase gene was used as the reference gene. The obtained Cq values were used as the original data to calculate the relative expression levels of DEGs via the 2^−ΔΔcq^ method.

### 2.10. Statistical Analysis

All data analyses were conducted with Data Processing System (DPS) statistical software (DPS 10.05, Hangzhou, Zhejiang, China), and the least-significant-different (LSD) test (*p* ≤ 0.05) was used to compare significant differences between treatments.

## 3. Results

### 3.1. Plant Growth

No significant difference in the net increase in shoot FW of the wheat seedlings was observed between CK, AC, and SA treatments at 24, 48, 72, and 96 h (Figure 1a,c). At 24 h, no significant difference in the net increase in root FW was observed between CK, SA, and AC treatments (Figure 1b). However, the net increase in root biomass of the SA-treated wheat seedlings was significantly decreased at 48, 72, and 96 h compared to CK, while the net increase under AC treatment was increased compared to SA (Figure 1b,c). These results indicate that NH_4_^+^ treatment suppresses root growth and the addition of HCO_3_^−^ partially restores root growth.

### 3.2. DEGs under Different N Treatments

After filtering out low-quality reads and adapter sequences, a total of 1.38 billion clean reads were obtained with a Q30-based percentage of 91.16% and an average GC content above 55.4%. Combining the GC content and Q30, we believed that the sequencing results were highly accurate and relatively reliable for further experimental analysis.

We performed differential significance analysis of DEGs from the roots of three plant groups (CK, SA, and AC) and conducted paired comparisons between wheat roots subjected to different N treatments. Our screening criteria for significance of DEGs were |log_2_ (fold change)| ≥ 1 and *p* ≤ 0.05. In the comparison between SA and CK, a total of 97,506 DEGs were identified, of which 5738 DEGs were upregulated and 6449 DEGs were downregulated. In AC vs. CK, 96,653 DEGs were identified, including 369 upregulated and 81 downregulated DEGs. In AC vs. SA, 97,383 DEGs were identified, with 10,507 upregulated and 9692 downregulated DEGs (Figure 2). These data suggest that the roots strongly respond to NH_4_^+^ stress and that the HCO_3_^−^-dependent alleviation of NH_4_^+^ stress is extremely complex.

### 3.3. Expression of Fermentation Genes and Concentration of Fermentation Products

The expression of fermentation transcripts, PDC, ADH, LDH, alanine aminotransferase (AlaAT), and NAD-dependent formate dehydrogenase (FDH) was upregulated under SA treatment at 48 h compared with CK. However, under AC treatment, the expression of these DEGs was significantly downregulated compared with SA treatment (Figure 3).

To further verify whether high NH_4_^+^ causes hypoxic stress in wheat roots, the contents of fermentation products were determined. The results show that NH_4_^+^ alone caused a significant accumulation of ethanol (62.95% increase compared to CK) in the roots (Figure 3). Although the root LA content was decreased in wheat plants under SA treatment, the LA content in the nutrient solution was increased by 23.20%, suggesting that a larger amount of LA was released from the root to apoplast and then to the medium. However, root ethanol accumulation and LA efflux rate were decreased by 39.36% and 10.44%, respectively, after the application of HCO_3_^−^ (Figure 3). These results suggest that NH_4_^+^ alone may induce low O_2_ stress in the root cells, leading to alcohol and lactate fermentation; the HCO_3_^−^-dependent alleviation of NH_4_^+^ stress may be associated with the relief of low O_2_ stress and the lower accumulation of fermentation products.

### 3.4. Expression of Genes Involved in Hypoxic Stress

To further verify whether SA treatment would induce hypoxic stress in terms of root growth, 11 classes of hypoxia-inducible genes were collected from previous publications (Table 1). Among these genes, hypoxia-inducible factor (HIF), burst oxidase homologs (Rbohs), internal and external alternative NADH dehydrogenase, alternative oxidases (AOXs), nodulin intrinsic proteins (NIPs), aspartate aminotransferase (AspAT), AlaAT, the pivotal enzymes of γ-aminobutyric acid (GABA shunt (glutamate decarboxylase (GAD), GABA transaminase (GABA-T), and succinate-semialdehyde dehydrogenase (SSADH)), ethylene biosynthesis, mitochondrial dicarboxylate carrier, mitochondrial arginine carrier, SLAH3, allene oxide synthase, and nudix hydrolase were significantly upregulated under NH_4_^+^ alone and downregulated after the application of HCO_3_^−^. On the contrary, prolyl 4-hydroxylases (PHDs), plasma membrane intrinsic proteins (PIPs), non-symbiotic hemoglobins (PGBs), DNA methylation, and chromatin structure regulatory mechanisms were significantly downregulated under NH_4_^+^ treatment and upregulated after the addition of HCO_3_^−^. The differential expression of these genes under SA is highly consistent with what was previously reported in plants under hypoxic stress (Table 1; for NIPs, see Figure 3), further suggesting that NH_4_^+^ treatment alone may induce hypoxic stress in wheat roots.

### 3.5. Expression of DEGs Involved in O_2_ Transport or Consumption Processes

To explore the possible causes of SA-induced hypoxic stress, we analyzed the transcript abundance of genes involved in O_2_ transport or consumption processes. The results show that the expressions of 14 genes encoding PIP-type AQPs were significantly downregulated and those of 5 genes encoding Rbohs were significantly upregulated under SA treatment (Table 1), and after the addition of HCO_3_^−^, the PIP AQPs were significantly upregulated and *Rbohs* genes were significantly downregulated (Table 1).

A total of 255 DEGs encoding dioxygenases were screened between N treatments. In the comparison between SA and CK, 47 DEGs were significantly upregulated and 13 DEGs were markedly downregulated (Figure 4). The upregulated dioxygenases are mainly involved in ethylene synthesis (1-aminocyclopropane-1-carboxylate oxidase 1 (ACO)), ABA synthesis (9-cis-epoxycarotenoid dioxygenase 1), auxin oxidation (2-oxoglutarate-dependent dioxygenase (DAO)), gibberellin 2-beta-hydroxylation (gibberellin 2-beta-dioxygenase 2), and fatty acid desaturation (fatty acid dioxygenase AlphaDOX1), while the downregulated DEGs are mainly involved in iron (Fe) transport (2′-deoxymugineic-acid 2′-dioxygenases). However, in AC vs. SA, 19 DEGs were significantly upregulated and 67 DEGs were significantly downregulated (Figure 4). Generally, these genes showed the opposite expression pattern to that in SA vs. CK. In short, the majority of DEGs encoding dioxygenases were significantly upregulated in the roots with NH_4_^+^ treatment alone, while most DEGs were generally downregulated after the addition of external HCO_3_^−^ (Figure 4).

### 3.6. Glycolysis, Pyruvate Metabolism, TCA Cycle, Fermentation, Shikimate Pathway, and GABA Shunt

The expression levels of DEGs involved in glycolysis, including phosphofructokinase (PFK), orthologs to phosphoglycerate kinase (PGK), and enolase (ENO), were generally upregulated in roots under SA treatment compared to CK, except for hexokinase (HK) (Figure 5). After the addition of HCO_3_^−^, the expression of DEGs encoding glycolytic enzymes, including PFK, PGK, and ENO, was downregulated, while HK was upregulated (Figure 5). These results indicate that the flux of glycolysis may be increased under NH_4_^+^ treatment and downregulated with the addition of HCO_3_^−^.

As shown in Figure 5, SA enhanced the transcript levels of DEGs encoding phosphoenolpyruvate carboxylase (PEPC), phosphoenolpyruvate carboxylase kinase (PEPCK), and NADP-malic enzyme (ME) for anaplerotic routes associated with the TCA cycle, while AC decreased the transcript levels of these genes. In the pyruvate metabolism and TCA cycle pathways, the expression of DEGs encoding PK, PDHC, malate dehydrogenase (MDH), and citrate synthase (CS) was downregulated under NH_4_^+^ treatment. After the addition of HCO_3_^−^, the expression of DEGs encoding PK, PDHC, MDH, and CS was upregulated (Figure 5). Briefly, these data indicate that the capacity of the TCA cycle may be suppressed in roots when wheat plants are exposed to NH_4_^+^ alone and may be promoted when HCO_3_^−^ is added.

The physiological assay showed that the activity of PDC, ADH, and LDH in roots was significantly increased by 36.69%, 43.66%, and 61.60%, respectively, under SA treatment and decreased by 19.69%, 13.52%, and 32.57% after the addition of HCO_3_^−^, whereas the activity of PK and pyruvate dehydrogenase (PDH) in roots decreased by 11.29% and 11.15%, respectively, under NH_4_^+^ treatment and increased by 4.37% and 93.17% after the addition of HCO_3_^−^ (Table 2), which was highly consistent with the RNA-Seq results (Figure 5).

The concentrations of Ala, formate, aromatic amino acids (Trp, Tyr, and Phe), and GABA increased significantly in the roots of wheat plants under SA treatment compared with CK. The addition of HCO_3_^−^ led to the decreased synthesis of these amino acids and formate. The concentrations of TCA cycle intermediates, including Pyr, CA, KGA, succinate, fumarate, malate, and OAA, were significantly decreased in the roots when subjected to SA treatment, while under AC treatment, the concentrations were increased (Table 3).

### 3.7. ATP Synthesis

All DEGs encoding ATP synthases were downregulated under SA treatment and upregulated when HCO_3_^−^ was added (Figure 6a). Accordingly, the activity of complex V significantly decreased in the roots under SA compared with CK and significantly increased under AC compared with SA (Figure 6b). As a consequence, root ATP content decreased and ADP content increased, resulting in a lower ATP/ADP ratio under SA treatment compared with CK (Figure 6c,d). Adding HCO_3_^−^ led to a significant increase in ATP content and decrease in ADP content, resulting in a higher ATP/ADP ratio compared with SA (Figure 6c,d).

### 3.8. DEGs Involved in Cell Division and Elongation

A total of 72 DEGs, including 40 genes encoding mitosis-related proteins, 26 genes associated with cell division, and 6 genes involved in cell elongationin the roots, were downregulated by SA treatmentand upregulated by AC treatment (Table 4). Taken together, these data suggest that NH_4_^+^ alone can suppress the rate of cell division in meristems and cell elongation in the elongation zone, thereby decreasing root growth, while adding HCO_3_^−^ promotes the cell cycle rate and restores root growth.

### 3.9. Functional Analysis of DEGs

Using GO classification, DEGs were classified into three functional categories: BP, CC, and MF. GO analysis showed that in SA vs. CK, the main categories in the BP group included microtubule-based movement, cell wall biogenesis, chromosome condensation, the mitotic cell cycle, DNA replication initiation, and pre-replicative complex assembly that is involved in nuclear cell cycle DNA replication. The DEGs in the CC group were mainly classified into the cell wall, microtubule, DNA replication preinitiation complex, and chromosome. The DEGs in the MF group were classified as electron carrier activity, DNA replication origin binding, the structural constituent of the cytoskeleton, nucleosomal DNA binding, DNA helicase activity, and glutamate-ammonia ligase activity (Figure 7a). In AC vs. SA, the main enriched DEGs in the BP group were associated with microtubule-based movement, cell wall biogenesis, the mitotic cell cycle, chromosome condensation, the aromatic compound biosynthetic process, and the response to auxin. The DEGs in the CC group mainly included the microtubule, cell wall, membrane, chromosome, DNA replication preinitiation complex, and mitochondrion. The DEGs in the MF group were mainly classified as GTPase activity, ATP-dependent microtubule motor activity, electron carrier activity, DNA replication origin binding, nucleosomal DNA binding, DNA helicase activity, transmembrane transporter activity, and glutathione oxidoreductase activity (Figure 7b).

KEGG analysis revealed that in SA vs. CK, the significantly enriched pathways included DNA replication; amino acid biosynthesis; alanine, aspartate, and glutamate metabolism; amino sugar and nucleotide sugar metabolism; nitrogen metabolism; glycolysis/gluconeogenesis; starch and sucrose metabolism; carbon metabolism; and the citrate cycle (TCA cycle) (Figure 8a). In AC vs. SA, the pathways were mainly clustered in DNA replication; amino acid biosynthesis; carbon metabolism; glycolysis/gluconeogenesis; the pentose phosphate pathway; nitrogen metabolism; alanine, aspartate, and glutamate metabolism; amino sugar and nucleotide sugar metabolism; nucleotide metabolism; the citrate cycle (TCA cycle); beta-alanine metabolism; and pyruvate metabolism (Figure 8b). These results indicate that an array of physiological processes in wheat roots are affected by NH_4_^+^ stress.

### 3.10. Validation of Hub Genes by qRT-PCR

Sixteen hub genes were selected to further verify the reliability of the RNA-Seq data by determining the RNA expression levels. The qRT-PCR results showed that the 7 hub genes, including ADH, 1-aminocyclopropane-1-carboxylate synthase (ACS), and AOXs, were significantly upregulated, and the 9 hub genes, associated with PIP, PDHC, DNA, and chromatin metabolic processes and mitosis, were significantly downregulated in SA vs. CK, while in AC vs. SA, these 16 DEGs showed the reverse expression trend. The qRT-PCR results were highly consistent with those of RNA-Seq analysis (Figure 9).

## 4. Discussion

### 4.1. HCO_3_^−^ Alleviates the Inhibition of Root Growth under NH_4_^+^ Treatment Alone

Increasing evidence has demonstrated that NH_4_^+^ as a dominant N source inhibits root growth in *Arabidopsis thaliana*, wheat, rice, and other plants [6,7,14,66], suggesting that roots are highly sensitive to NH_4_^+^ [25]. In the present study, we found that NH_4_^+^ treatment alone inhibited root growth and adding HCO_3_^−^ attenuated the inhibitory effects of NH_4_^+^ (Figure 1a–c).

### 4.2. Fermentation Is Stimulated by NH_4_^+^ and Mitigated after Addition of HCO_3_^−^

The underlying mechanisms of NH_4_^+^ toxicity remain largely unknown [7,21,26]. In the present study, we found an increased transcript abundance of DEGs encoding PDC, ADH, and LDH; an increased activity of PDC, ADH, and LDH; and subsequent ethanol and LA accumulation under NH_4_^+^ treatment, while the addition of HCO_3_^−^ significantly attenuated these changes (Figure 3 and Figure 9, Table 2). The increased rate of LA efflux under NH_4_^+^ treatment may be due to the higher expression of NIPs (Figure 3), as observed in *Arabidopsis* and other plants under hypoxic conditions [67,68]. These results are highly consistent with the findings reported in *Glycine max* [69]. Considering that PDC, ADH, and LDH are reliable markers of fermentative processes launched by hypoxic stress [49,51,60], it would be reasonable to assume that NH_4_^+^ alone can induce hypoxia in the roots and exogenous HCO_3_^−^ can attenuate this stress. This assumption corroborates the finding that the plant response to NH_4_^+^ may overlap with the response to low O_2_ stress [32].

### 4.3. Differential Expression of Hypoxia Response Genes Indicates That NH_4_^+^ Induces Cellular O_2_ Deprivation and HCO_3_^−^ Alleviates This Stress

Previous studies have reported that numerous hypoxia-inducible factors, such as HIF, Rbohs, internal and external alternative NADH dehydrogenases, AOXs, NIPs, AlaAT, and ethylene biosynthesis, were significantly induced by hypoxic stress, while PIPs, PGB, PHDs, DNA methylation, and chromatin structure regulatory mechanisms were significantly downregulated. In our study, we observed that the transcriptomic response of core hypoxia-inducible genes to NH_4_^+^ treatment was highly consistent with what was observed in earlier studies on hypoxia-stressed plants (Table 1, Figure 3), further supporting the notion that NH_4_^+^ treatment may induce hypoxic stress in wheat roots. The HCO_3_^−^-dependent alleviation of NH_4_^+^ toxicity may be associated with the attenuated hypoxic stress.

### 4.4. O_2_ Uptake, Transport, and Consumption May Be Associated with Cellular O_2_ Availability

It is interesting to explore how hypoxic conditions are established in wheat roots under NH_4_^+^ treatment. Rbohs are a family of plasma-membrane-bound enzymes that transfer electrons from cytosolic NADPH/NADH to apoplastic O_2_ with the production of reactive oxygen species, and thus play various roles in defense response and morphogenetic processes [12,43]. In this study, *Rbohs* genes were significantly upregulated in roots subjected to NH_4_^+^ and significantly downregulated after the application of HCO_3_^−^ (Table 1). These data suggest that the higher expression of *Rbohs* genes under NH_4_^+^ treatment dampens O_2_ uptake and contributes to cellular O_2_ depletion (Figure 10).

Plant AQPs are localized in cell membranes to transport water molecules, O_2_, and CO_2_, and are involved in the hypoxia response [72,73,74]. Five subfamilies of AQPs, including the PIPs and NIPs, have been categorized in higher plants [75]. In particular, the overexpression of PIP1;3 has been observed to improve the rate of root O_2_ utilization and respiration, and to promote plant growth under hypoxic stress by mediating glycolysis, pyruvate metabolism, and the TCA cycle in the roots of canola (*Brassica napus*) [76]. In the present study, we found that NH_4_^+^ treatment reduced the transcript levels of genes encoding PIP-type AQPs, while the addition of HCO_3_^−^ led to a significantly upregulated expression (Table 1, Figure 9). Based on these results, it is conceivable that the downregulated expression of PIP-type AQPs at least partially contributes to the low O_2_ stress in roots under NH_4_^+^ treatment (Figure 10).

Dioxygenases catalyze the incorporation of one or two O_2_ atoms into target organic substrates in various metabolic reactions, including DNA replication, RNA modification, and histone demethylation [41,77]. Therefore, high dioxygenase activity would induce O_2_ overconsumption [41]. In this study, we observed that most genes encoding dioxygenases, including ACO1, non-heme dioxygenases, gibberellin 2-beta-dioxygenases, and 9-cis-epoxycarotenoid dioxygenases, were upregulated by NH_4_^+^ treatment (Figure 4). So, we speculate that the higher upregulated expression of dioxygenases would promote the incorporation of molecular O_2_ into various substrates and decrease free O_2_ availability, thus leading to cellular hypoxia in the roots of plants fed NH_4_^+^ alone (Figure 10).

AOX is one of the terminal oxidases of the plant mitochondrial ETC [46,78]. AOX has a non-proton motive characteristic and delivers electrons from ubiquinone to O_2_ to generate H_2_O by bypassing two sites of H^+^ pumping in complexes III and IV of the cytochrome pathway, which dramatically reduces ATP generation [44,46,78]. Our results show that the transcript abundance of four AOX genes was increased in the roots of plants fed only NH_4_^+^, while HCO_3_^−^ supplementation led to a significantly decreased expression of these genes (Table 1, Figure 9). The upregulation of AOX genes under NH_4_^+^ treatment may greatly increase O_2_ consumption without ATP production, which may be associated with hypoxic stress in root cells (Figure 10).

PGBs have an extremely high affinity for O_2_ and extremely slow O_2_ dissociation properties [79] and can serve as terminal electron acceptors in hypoxic root tissue [80,81]. Under hypoxic conditions, the expression of *Pgb1.1* and *Pgb1.2* was upregulated, thus mitigating the inhibitory effect of O_2_ deprivation on root growth in maize [43]. Fe is involved in the biosynthesis of heme molecules [79], and its deficiency leads to physiological hypoxia [82]. Mugineic acid (MA) is involved in Fe translocation in plant tissues as an Fe chelator, and its biosynthesis requires a precursor, nicotianamine (NA, formed from S-adenosyl methionine via nicotianamine synthase (NAS)) in graminaceous plants [83] and 2′-deoxymugineic-acid 2′-dioxygenases [84]. Our results show that all genes encoding PGBs, NAS, 2′-deoxymugineic-acid 2′-dioxygenases, and Fe^2+^ transport proteins in the roots were significantly downregulated under NH_4_^+^ treatment and were upregulated after HCO_3_^−^ application (Table 1). Considering that Fe is a cofactor of PGBs, and PGBs function as O_2_ carriers and potential terminal electron acceptors in hypoxic root tissue, it is tempting to speculate that intracellular O_2_ availability and respiratory use are reduced in the roots of wheat plants fed NH_4_^+^ and increased after the addition of HCO_3_^−^ (Figure 10).

### 4.5. Glycolytic Pathway Is Stimulated by NH_4_^+^ but Mitigated by Supplementation with HCO_3_^−^

It has been well documented that O_2_ is used as a terminal electron acceptor in ETC [46]. During aerobic respiration, about 36 moles of ATPs are produced [85]. However, under hypoxic conditions, ATP synthesis from oxidative phosphorylation is reduced [41]; thus, plant cells will rely on other metabolic pathways, such as glycolysis, leading to reduced ATP generation (2 moles of ATPs per mole of glucose) [52,85]. This metabolic switch may be regulated by many factors, such as hypoxia-inducible factor 1 (HIF-1) [85]. In the present study, we observed that DEGs encoding PFK, PGK, and ENO were upregulated under NH_4_^+^ treatment and downregulated after HCO_3_^−^ was added (Figure 5). In short, NH_4_^+^ treatment simulates glycolysis, presumably because of the lower O_2_ availability in the root cells, while the addition of HCO_3_^−^ may mitigate the hypoxic stress and improve the glycolytic pathway.

### 4.6. Supplementing with HCO_3_^−^ Ameliorates NH_4_^+^-Repressed TCA Cycle

It is well known that the TCA cycle provides essential C skeletons for the assimilation of NH_4_^+^ into amino acids [34], while producing energy [5]. Therefore, the functions of the TCA cycle may be important in the alleviation of NH_4_^+^ toxicity. In the present study, we found that the DEGs encoding PEPC and PEPCK were upregulated under NH_4_^+^ treatment and downregulated after HCO_3_^−^ was added (Figure 5). PEPC catalyzes the carboxylation of PEP in the presence of HCO_3_^−^ to form OAA, and PEPCK catalyzes the decarboxylation of OAA to PEP in the gluconeogenesis pathway [86]. Therefore, the higher expression of PEPC and PEPCK under NH_4_^+^ conditions could lead to futile cycling between PEP and OAA in the cytoplasm, resulting in a lower accumulation of OAA in the roots (Table 3), which decreases the C anaplerosis in the TCA cycle.

In addition, PEP can also be converted to chorismate through the shikimate pathway, producing aromatic amino acids Trp, Tyr, and Phe [87]. In this study, we observed that under NH_4_^+^ conditions, the expression of all *AroL*, *AroC*, and *PheA* genes encoding the shikimate pathway enzymes was significantly upregulated and the concentrations of Trp, Tyr, and Phe were increased, which in turn significantly reduced the concentration of Pyr (glycolytic terminal intermediate) under NH_4_^+^ treatment (Table 1 and Table 3). The application of HCO_3_^−^ led to increased Pyr concentration (Table 3).

Ala and formate are synthesized via AlaAT and FDH, respectively, using Pyr as a precursor [88,89]. *AlaAT* and *FDH* expression was induced and Ala and formate accumulated in roots under NH_4_^+^ treatment, and these were repressed by the addition of HCO_3_^−^ (Figure 3), leading to more ions in the NH_4_^+^-treated plants (Table 3). Furthermore, the expression levels and activity of PK and PDHC decreased with NH_4_^+^ treatment and increased after the addition of HCO_3_^−^ (Figure 5 and Figure 9, Table 2). These changes were greatly attenuated after the addition of HCO_3_^−^. Based on these results, it can be reasonably hypothesized that NH_4_^+^ nutrition alone suppresses the flux of Pyr in the TCA cycle and adding HCO_3_^−^ mitigates this suppression.

The irreversible *α*-decarboxylation of glutamate catalyzed by GAD in plant tissues synthesizes GABA, a bypassing step in the TCA cycle known as GABA shunt [90]. In our study, the expression of *GDA*, *GABA-T*, and *SSADH* genes and the GABA content were significantly upregulated in roots under NH_4_^+^ treatment, while after HCO_3_^−^ addition, the expression was downregulated and GABA content was decreased (Table 1 and Table 3). Furthermore, we observed that the expression of most TCA cycle enzymes and the concentrations of key intermediates were reduced under NH_4_^+^ treatment and increased after HCO_3_^−^ was applied (Figure 5, Table 3). Based on these results, we propose that NH_4_^+^ treatment alone reduces the overall capacity of the TCA cycle, likely due to the suppressed flow of C and enhanced GABA shunt. It was encouraging to find that the addition of HCO_3_^−^ mitigated the suppressed TCA cycle activity. This finding is consistent with findings reported in wheat [53] and rice [91] and findings reported by Cramer and Lewis [30] and Bialczyk et al. [34].

### 4.7. ATP Biosynthesis Is Inhibited by NH_4_^+^ and Promoted after Addition of HCO_3_^−^

As discussed above, the functions of alternative NAD(P)H:ubiquinone oxidoreductases and AOXs are not coupled to proton translocation, and thus inhibit ATP production [46,92]. In the present study, we found that the expression of alternative NAD(P)H:ubiquinone oxidoreductases and AOXs was upregulated in roots under NH_4_^+^ treatment and downregulated after the addition of HCO_3_^−^ (Table 1, Figure 9), while the transcription level of *PGBs* was reduced under NH_4_^+^ alone and increased after the addition of HCO_3_^−^ (Table 1). In agreement with this, we observed that the downregulated ATP synthase expression, reduced ETC complex V activity, and lower ATP content under NH_4_^+^ treatment were largely mitigated after the addition of HCO_3_^−^ (Figure 6a–c). Considering that hypoxic stress induces the fermentation and glycolysis pathways, suppresses the TCA cycle, and limits energy generation [85,93], we conclude that NH_4_^+^-induced changes in the roots may be due to low O_2_ stress, and that adding HCO_3_^−^ may promote O_2_ transport to the mitochondria and oxidative phosphorylation with PGBs as electron acceptors, ultimately improving ATP synthesis in the roots.

### 4.8. HCO_3_^−^ Mitigates NH_4_^+^-Induced Cell Cycle Arrest and Elongation Inhibition

Cell division and cell elongation are the principal processes that determine root growth, and they occur in spatially distinct developmental zones [94]. The precise regulation of chromatin structure in the nucleus is closely related to the transcriptional reprogramming associated with cell proliferation in root apical meristems and growth and development in *Arabidopsis* and rice [95]. In this study, we found that the expression of all genes related to DNA and chromatin metabolic processes, cell division, and cell elongation was downregulated in roots exposed to NH_4_^+^ and upregulated after the addition of HCO_3_^−^ (Table 1 and Table 4, Figure 9). Hence, we propose that cell cycle arrest and elongation inhibition may directly account for root growth inhibition under NH_4_^+^ treatment and that applying HCO_3_^−^ attenuates the inhibitory effects of this treatment.

### 4.9. Ethylene Signaling Is Involved in Regulating NH_4_^+^ Toxicity and its Alleviation by HCO_3_^−^

Numerous studies have shown that ethylene contributes to the expression of core hypoxia genes and hypoxia acclimation by enhancing the production and stabilization of Group VII Ethylene Response Factors (ERFVIIs) [66,96,97]. ERFVIIs facilitate the induction of genes involved in fermentation, glycolysis, energy maintenance, and C metabolism under hypoxic stress in *Arabidopsis* [98], rice [99], and tobacco [44]. In the present study, transcriptome-wide analysis showed that the expression of genes encoding ACSs, ACOs, and two pivotal enzymes in ethylene synthesis was upregulated under NH_4_^+^ treatment but downregulated after HCO_3_^−^ addition (Table 1, Figure 9). It seems reasonable to assume that ethylene synthesis is increased and is involved in NH_4_^+^ toxicity as a signaling agent, and that adding HCO_3_^−^ negatively regulates ethylene signal transduction, thus improving root growth (Figure 10).

## 5. Conclusions

Treatment with NH_4_^+^ alone led to significantly restrained wheat root growth and induced the expression of hypoxia-response core genes, indicating that hypoxia may be the primary cause of NH_4_^+^ toxicity in root cells. Hypoxic stress appears to be associated with the upregulation of Rbohs, dioxygenases, and AOXs and the downregulation of AQPs. As a consequence, the capacity of the TCA cycle is reduced and the production of ATP is inhibited, eventually dampening the root cell cycle, elongation, and growth. Compared with NH_4_^+^ nutrition alone, the addition of HCO_3_^−^ significantly improved hypoxia-related metabolic processes, boosted ATP generation, promoted root cell division and elongation, and ultimately enhanced root growth in wheat seedlings. Ethylene signaling may be involved in NH_4_^+^ toxicity and HCO_3_^−^-dependent detoxification.

As the first report on hypoxic stress triggered by NH_4_^+^ treatment, this study provides novel insights into the mechanisms of NH_4_^+^ toxicity and its alleviation, further providing a valuable molecular basis for studying how to improve NUE. In particular, based on the data reported here, we can strongly recommend splitting the proportion of N fertilizer for wheat production in the field. Alternatively, C-containing fertilizers such as ammonium bicarbonate and urea should be used as the preferred N source. Retaining the C-rich residue of previous crops is also a practical strategy when using NH_4_^+^ fertilizers as the dominant N source. These practices could minimize NH_4_^+^ toxicity and increase nutrient use efficiency and wheat grain yield.

## Figures and Tables

**Figure 1 biology-13-00101-f001:**
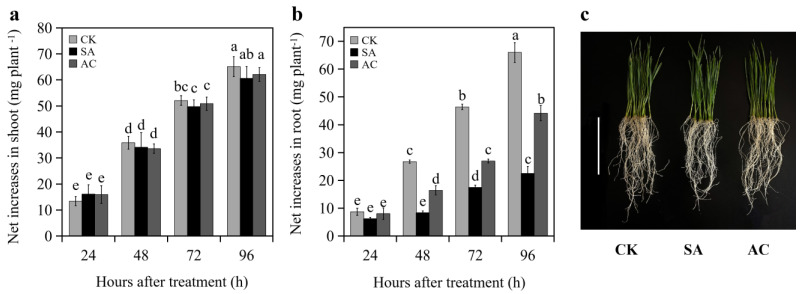
Net increase in (**a**) shoot and (**b**) root fresh weight (mean ± SD) of wheat seedlings at 24, 48, 72, and 96 h after treatment, and (**c**) seedling phenotype grown with different treatments at 48 h. Different letters above columns indicate significant differences between treatments at *p* ≤ 0.05. In (**c**), bar = 10 cm. Wheat seedlings were treated with 7.5 mM NO_3_^−^ (CK), 7.5 mM NH_4_^+^ (SA), or 7.5 mM NH_4_^+^ + 3 mM HCO_3_^−^ (AC).

**Figure 2 biology-13-00101-f002:**
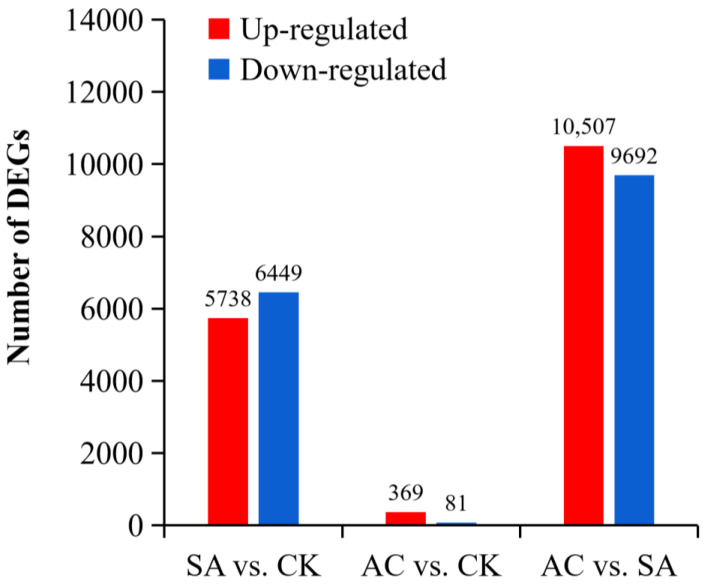
Paired comparisons of DEGs between N treatments. Red and blue represent up- and downregulated expression, respectively. Wheat seedlings were treated with 7.5 mM NO_3_^−^ (CK), 7.5 mM NH_4_^+^ (SA), or 7.5 mM NH_4_^+^ + 3 mM HCO_3_^−^ (AC).

**Figure 3 biology-13-00101-f003:**
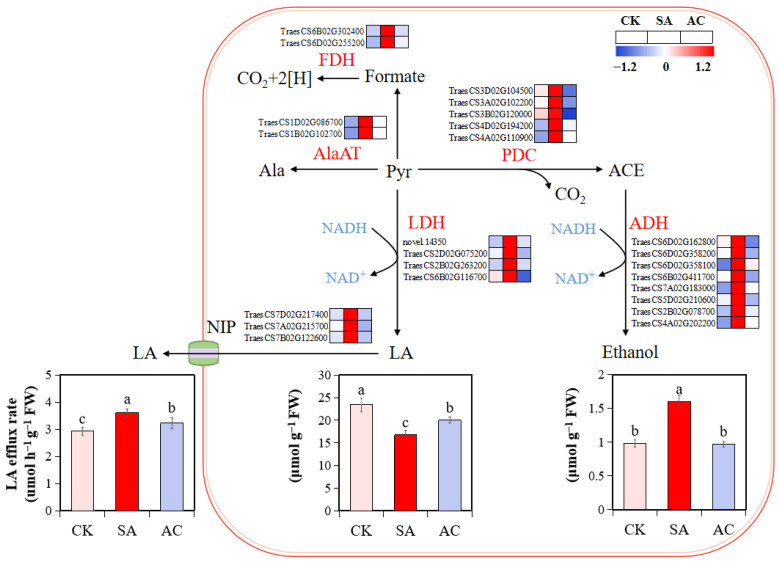
Influence of N treatments on fermentation pathway. Heatmap shows expression levels of identified DEGs in roots of wheat plants with different treatments. Numbers in color scale bar indicate log_2_ (FC) in gene expression. Wheat seedlings were treated with 7.5 mM NO_3_^−^ (CK), 7.5 mM NH_4_^+^ (SA), or 7.5 mM NH_4_^+^ + 3 mM HCO_3_^−^ (AC). Different letters above bars indicate significant differences at *p* ≤ 0.05. ACE, acetaldehyde; ADH, alcohol dehydrogenase; Ala, alanine; AlaAT, alanine aminotransferase; FDH, formate dehydrogenase; LA, lactate; LDH, lactate dehydrogenase; NIP, nodulin intrinsic proteins; PDC, pyruvate decarboxylase; Pyr, pyruvate.

**Figure 4 biology-13-00101-f004:**
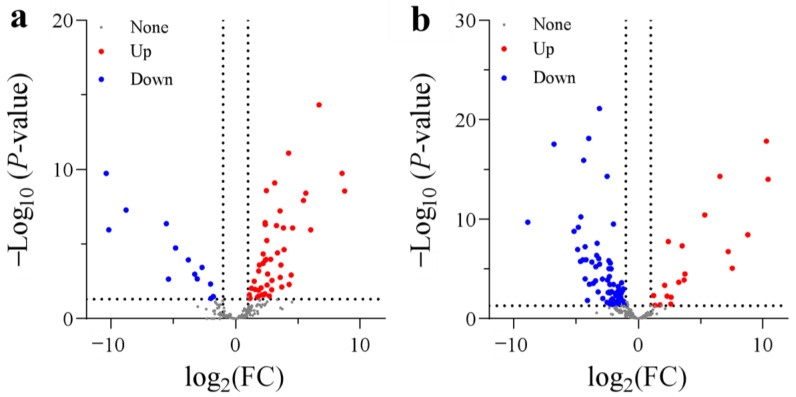
Volcano plots of DEGs encoding dioxygenases: (**a**) SA vs. CK; (**b**) AC vs. SA. Wheat seedlings were treated with 7.5 mM NO_3_^−^ (CK), 7.5 mM NH_4_^+^ (SA), or 7.5 mM NH_4_^+^ + 3 mM HCO_3_^−^ (AC). Red and blue dots indicate upregulated and downregulated DEGs, respectively, and gray dots indicate genes that were not differentially expressed. Images were created using GraphPad Prism 9.4.1.

**Figure 5 biology-13-00101-f005:**
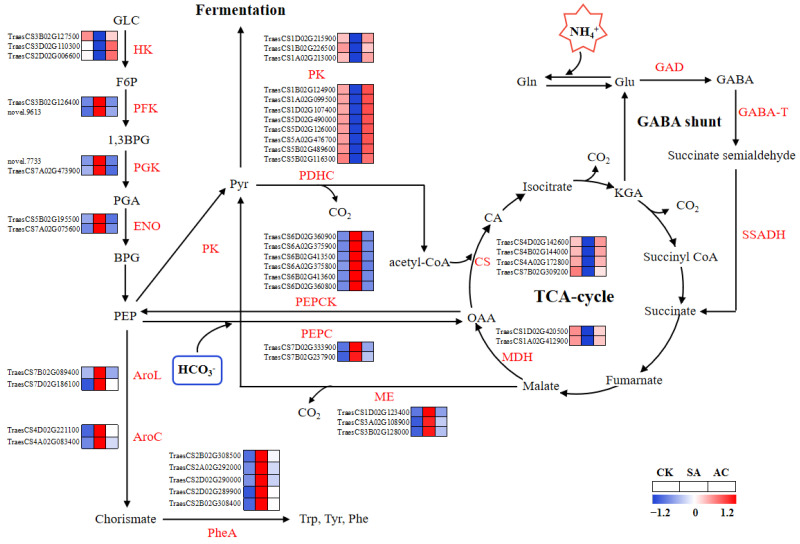
Glycolysis and TCA cycle pathways overrepresented among differentially expressed genes and significantly changed metabolites. Color gradient indicates expression levels of DEGs from low (blue) to high (red). Wheat seedlings were treated with7.5 mM NO_3_^−^ (CK), 7.5 mM NH_4_^+^ (SA), or 7.5 mM NH_4_^+^ + 3 mM HCO_3_^−^ (AC). AroC, chorismate synthetase; AroL, shikimate kinase; BPG, bisphosphoglycerate; 1,3 BPG, 1,3-bisphosphoglycerate; CA, citrate; CS, citrate synthase; ENO, enolase; F6P, fructose-6-phosphate; GABA, γ-aminobutyric acid; GABA-T, GABA transaminase; GAD, glutamate decarboxylase; GLC, glucose; Gln, glutamine; Glu, glutamate; HK, hexokinase; KGA, α-ketoglutarate; MDH, malate dehydrogenase; ME, malic enzyme; OAA, oxaloacetic acid; PDHC, pyruvate dehydrogenase complex; PEP, phosphoenolpyruvate; PEPC, phosphoenolpyruvate carboxylase; PEPCK, phosphoenolpyruvate carboxylase kinase; PFK, phosphofructokinase; PGA, 3-phosphoglycerate; PGK, phosphoglycerate kinase; Phe, phenylalanine; PheA, prephenate dehydratase; PK, pyruvate kinase; Pyr, pyruvate; SSADH, succinate-semialdehyde dehydrogenase; Trp, tryptophan; Tyr, tyrosine.

**Figure 6 biology-13-00101-f006:**
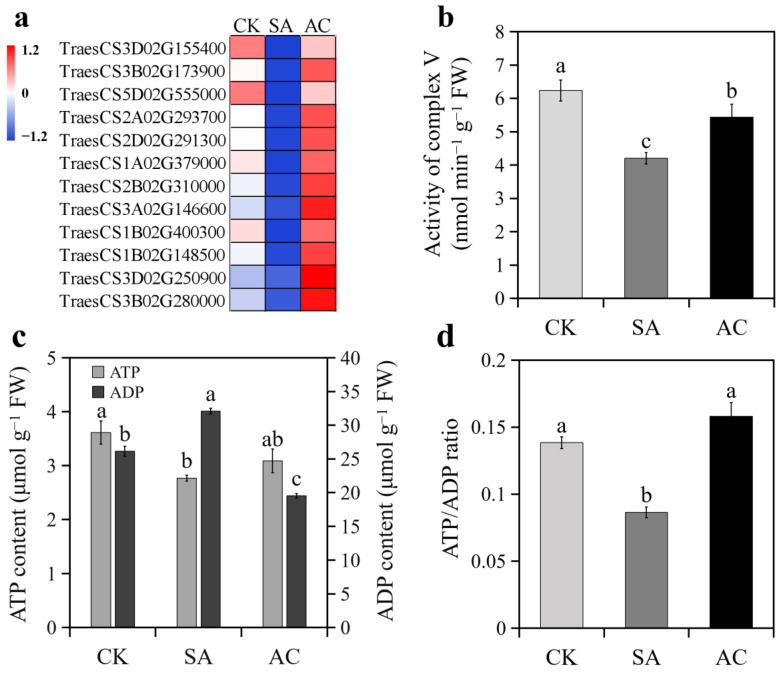
Effects of N treatments on ATP synthesis in roots of wheat seedlings: (**a**) expression of DEGs encoding ATP synthases; (**b**) activity of root complex V; (**c**) ATP and ADP content; (**d**) ratio of ATP to ADP. Values represent mean ± SD from three independent biological replicates. Different lowercase letters above columns indicate significant differences at *p* < 0.05. Wheat seedlings were treated with 7.5 mM NO_3_^−^ (CK), 7.5 mM NH_4_^+^ (SA), or 7.5 mM NH_4_^+^ + 3 mM HCO_3_^−^ (AC).

**Figure 7 biology-13-00101-f007:**
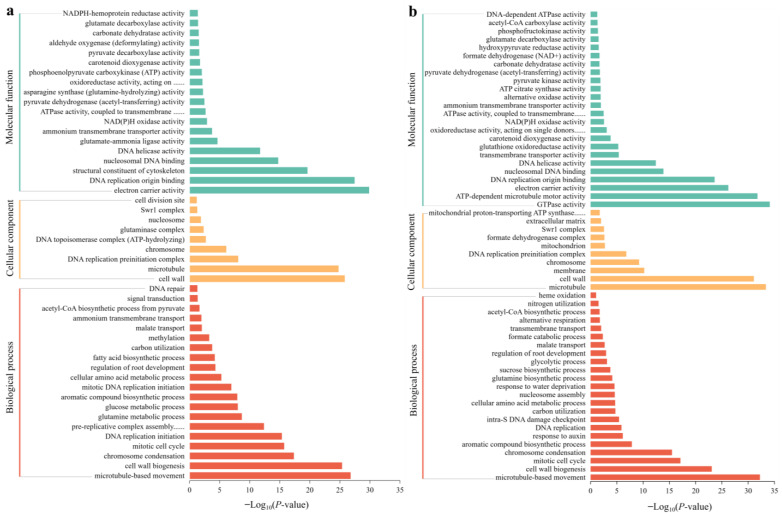
GO analysis of DEGs under different N treatments: (**a**) SA vs. CK; (**b**) AC vs. SA. *X*-axis indicates −Log_10_ (*p*-value), and *Y*-axis is enriched GO terms.

**Figure 8 biology-13-00101-f008:**
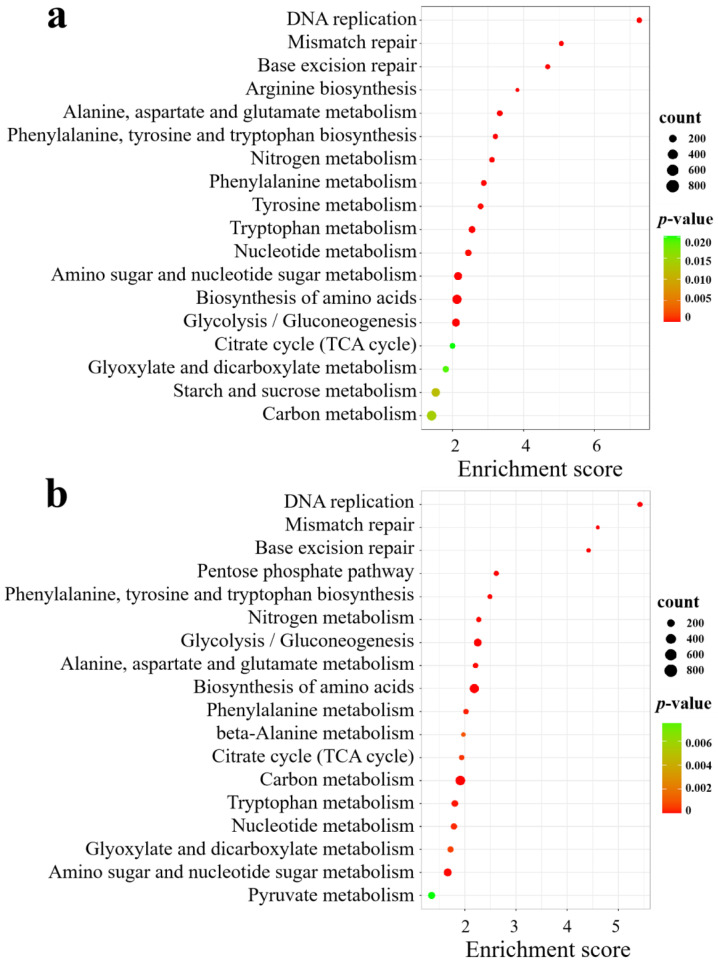
KEGG analysis of DEGs under different N treatments: (**a**) SA vs. CK; (**b**) AC vs. SA. *X*-axis is enrichment score, *Y*-axis is KEGG pathway; size of dots represents number of genes annotated to KEGG pathway; and red to green in color bar indicate significance (high to low, respectively) of enrichment.

**Figure 9 biology-13-00101-f009:**
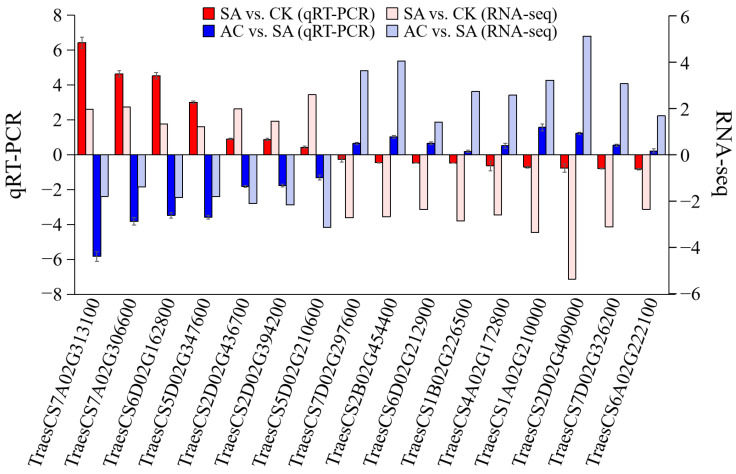
RNA-Seq data accuracy verification. Column represents 2^−ΔΔcq^ value of qRT-PCR analysis and log_2_(FC) value of RNA-Seq. Values represent mean ± SD. Positive values on *Y*-axis indicate upregulation of hub genes, and negative values indicate downregulation. Wheat seedlings were treated with 7.5 mM NO_3_^−^ (CK), 7.5 mM NH_4_^+^ (SA), or 7.5 mM NH_4_^+^ + 3 mM HCO_3_^−^ (AC).

**Figure 10 biology-13-00101-f010:**
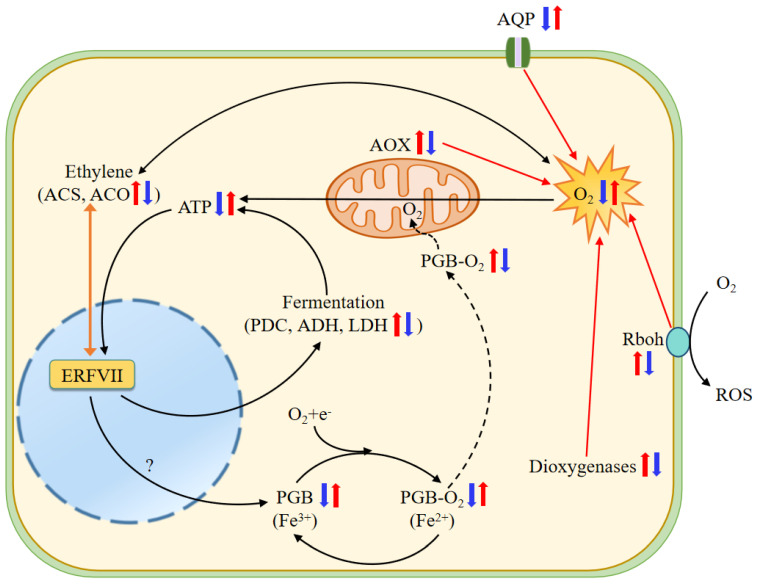
Schematic model of NH_4_^+^ toxicity and HCO_3_^−^-dependent alleviation in roots of wheat seedlings. Under NH_4_^+^ treatment, significantly upregulated Rbohs consume large amounts of apoplastic O_2_, downregulated PIP-type AQPs decrease O_2_ uptake and transport from the ambient environment to the cell, and upregulated dioxygenases and AOXs increase intracellular O_2_ consumption, thus reducing cellular O_2_ availability. O_2_ deprivation then suppresses oxidative phosphorylation and ATP production, in turn activating the ethylene-enhanced ERFVII pool and stimulating hypoxic metabolism, such as alcoholic and lactic fermentation, and regulates the expression of hypoxia-like responsive genes encoding PDC, ADH, LDH, and PGBs [70]. O_2_-binding PGBs may function as terminal electron acceptor electron transport chains (ETCs), thus downregulated expression of PGBs under NH_4_^+^ may reduce electron transport inETC and then ATP generation. Higher ethylene production further consumes more molecular O_2_ [71]. Conversely, adding HCO_3_^−^ greatly ameliorates the negative effects of NH_4_^+^ alone on these processes. Two juxtaposed arrows indicate differential expression of genes in SA vs. CK and AC vs. SA. The question mark in the figure indicates that this process remains to be elucidated. Red and blue arrows indicate increased and decreased transcript abundance, respectively. ACO, 1-aminocyclopropane-1-carboxylate oxidase; ACS, 1-aminocyclopropane-1-carboxylate synthase; ADH, alcohol dehydrogenase; AOX, alternative oxidase; AQP, aquaporins; ERFVII, Group VII Ethylene Response Factor; LDH, lactate dehydrogenase; PDC, pyruvate decarboxylase; PGB, nonsymbiotic hemoglobin; Rbohs, respiratory burst oxidase homolog.

**Table 1 biology-13-00101-t001:** List of genes related to hypoxic stress in wheat roots.

Gene ID	Gene Expression 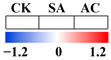			Gene Description	Species	References
*Hypoxia-induced proteins and regulators of hypoxia-inducible factor (HIF) pathway*						
TraesCS2D02G438000				Hypoxia-induced protein conserved region	Mammalia	[40]
TraesCS1B02G316600				Probable prolyl 4-hydroxylase 4	Mammalia	[41]
TraesCS6B02G456400				Probable prolyl 4-hydroxylase 3	Mammalia	[42]
*Burst oxidase homologs*						
TraesCS4D02G324800				Respiratory burst oxidase homolog protein C	*Zea mays*	[43]
TraesCS5D02G306400				Respiratory burst oxidase homolog protein C	*Nicotiana tabacum*	[44]
TraesCS5D02G105900				Respiratory burst oxidase homolog protein B		
TraesCS5A02G301700				Respiratory burst oxidase homolog protein C		
TraesCS4B02G327800				Respiratory burst oxidase homolog protein C		
*Alternative electron transport pathways*						
TraesCS3D02G343900				Internal alternative NADH dehydrogenase NDA1	*Zea mays*	[45]
TraesCS7A02G306600				External alternative NADH dehydrogenase NDB2		
TraesCS7D02G303500				External alternative NADH dehydrogenase NDB2		
TraesCS7B02G206900				External alternative NADH dehydrogenase NDB2		
TraesCS2A02G439400				Alternative oxidase 1a, mitochondrial	*Nicotiana tabacum*	[46]
TraesCS2D02G436700				Alternative oxidase 1a, mitochondrial	*Nicotiana tabacum*	[44]
TraesCS2A02G438200				Alternative oxidase 1b, mitochondrial		
TraesCS2B02G459300				Alternative oxidase 1a, mitochondrial		
*Aquaporin*						
TraesCS5D02G561700				Aquaporin PIP2-2	*Glycine max*	[47]
TraesCS7B02G002000				Probable aquaporin PIP2-1	*Sorghum bicolor*	[48]
TraesCS5B02G570800				Aquaporin PIP2-2		
TraesCS2B02G425600				Aquaporin PIP1-3/PIP1-4		
TraesCS6D02G212900				Probable aquaporin PIP2-2		
TraesCS6A02G222100				Probable aquaporin PIP2-2		
TraesCS2D02G404800				Aquaporin PIP1-3/PIP1-4		
TraesCS2A02G407700				Aquaporin PIP1-3/PIP1-4		
TraesCS2A02G065700				Probable aquaporin PIP2-6		
TraesCS6B02G259000				Probable aquaporin PIP2-2		
TraesCS6A02G405600				Aquaporin PIP1-5		
TraesCS2B02G077700				Probable aquaporin PIP2-6		
TraesCS2D02G063900				Probable aquaporin PIP2-6		
TraesCS5A02G181000				Aquaporin PIP1-5		
*N metabolism and GABA shunt*						
TraesCS4B02G053600				Glutamate decarboxylase 1	*Arabidopsis thaliana*	[49]
TraesCS4A02G261000				Glutamate decarboxylase 1	*Cucumis sativus*	[50]
TraesCS4D02G053600				Glutamate decarboxylase 1		
TraesCS1A02G374600				Glutamate decarboxylase		
TraesCS2D02G303600				Aspartate aminotransferase	*Cucumis sativus*	[50]
TraesCS2D02G451900				Aspartate aminotransferase, mitochondrial		
TraesCS2A02G452100				Glutamate/aspartate-prephenate aminotransferase		
TraesCS3B02G047500				Aspartate aminotransferase		
TraesCS2A02G188100				Glutamate/aspartate-prephenate aminotransferase		
TraesCS2B02G219100				Alanine aminotransferase 2	*Medicago truncatula*	[51]
TraesCS5A02G336500				Alanine aminotransferase 2	*Lotus japonicus*	[52]
TraesCS7D02G326200				Gamma-aminobutyrate transaminase 1, mitochondrial	*Oryza sativa*	[53]
TraesCS7A02G323200				Probable gamma-aminobutyrate transaminase 4		
TraesCS5B02G335500				Succinate-semialdehyde dehydrogenase, mitochondrial	*Arabidopsis thaliana*	[54]
TraesCS5D02G341200				Succinate-semialdehyde dehydrogenase, mitochondrial		
TraesCS7A02G329300				Succinate-semialdehyde dehydrogenase, mitochondrial		
*O*_2_-*binding and Fe transport*						
TraesCS1B02G350800				Non-symbiotic hemoglobin	*Oryza sativa*	[55]
TraesCS1A02G338400				Non-symbiotic hemoglobin 1		
TraesCS6D02G148200				Nicotianamine synthase 1	*Citrus junos*	[56]
TraesCS6A02G163200				Nicotianamine synthase 1		
TraesCS3B02G479500				2′-Deoxymugineic-acid 2′-dioxygenase	Mammalia	[42]
TraesCS3D02G437500				2′-Deoxymugineic-acid 2′-dioxygenase		
TraesCS3A02G445000				2′-Deoxymugineic-acid 2′-dioxygenase		
TraesCS4A02G294300				Fe(^2+^) transport protein 1		
TraesCS4B02G019300				Fe(^2+^) transport protein 1		
*DNA and chromatin metabolic processes*						
TraesCS4B02G053600				ATP-dependent DNA helicase DDM1	*Arabidopsis thaliana*	[57]
TraesCS4A02G261000				ATP-dependent DNA helicase DDM1		
TraesCS4D02G053600				ATP-dependent DNA helicase DDM1		
TraesCS1A02G374600				Increased DNA methylation 1		
TraesCS2D02G303600				Protein RNA-directed DNA methylation 12		
TraesCS2D02G451900				Protein RNA-directed DNA methylation 12		
TraesCS2A02G452100				ATP-dependent DNA helicase DDM1		
TraesCS3B02G047500				RuvB-like protein 1		
TraesCS2A02G188100				RuvB-like 2		
TraesCS2B02G219100				RuvB-like 2		
TraesCS5A02G336500				Chromatin assembly factor 1 subunit FSM	*Arabidopsis thaliana*	[58]
TraesCS7D02G326200				Nucleosome/chromatin assembly factor group D 07		
TraesCS7A02G323200				Chromatin assembly factor 1 subunit FAS2 homolog		
TraesCS5B02G335500				Chromatin assembly factor 1 subunit FSM		
TraesCS5D02G341200				Chromatin assembly factor 1 subunit FSM		
TraesCS7A02G329300				Nucleosome/chromatin assembly factor group D 07		
TraesCS7D02G319800				Chromatin assembly factor 1 subunit FAS2 homolog		
TraesCS7B02G230000				Nucleosome/chromatin assembly factor group D 07		
TraesCS2B02G466000				Nucleosome/chromatin assembly factor group D 06		
TraesCS2A02G445000				Nucleosome/chromatin assembly factor group D 06		
TraesCS5B02G332200				Histone-binding protein MSI1 homolog		
TraesCS7B02G224100				Chromatin assembly factor 1 subunit FAS2 homolog		
TraesCS5A02G331900				Histone-binding protein MSI1 homolog		
TraesCS5B02G403900				Protein TRI1		
TraesCS5D02G408400				Protein TRI1		
novel.8650				Replication Fork Protection Component Swi3		
novel.10308				Replication Fork Protection Component Swi3		
TraesCS2A02G072700				SWI/SNF-related matrix-associated actin-dependent regulator	Mammalia	[59]
TraesCS3D02G068900				Probable chromatin-remodeling complex ATPase chain		
TraesCS3B02G083400				Probable chromatin-remodeling complex ATPase chain		
TraesCS3A02G069900				Probable chromatin-remodeling complex ATPase chain		
*Ethylene signaling*						
TraesCS2A02G026800				1-Aminocyclopropane-1-carboxylate oxidase homolog 2	*Cucumis sativus*	[50]
TraesCS5D02G241100				1-Aminocyclopropane-1-carboxylate oxidase 1	*Nicotiana tabacum*	[44]
TraesCS2B02G040100				1-Aminocyclopropane-1-carboxylate oxidase homolog 1		
TraesCS6A02G325700				1-Aminocyclopropane-1-carboxylate oxidase 3		
TraesCS2A02G026500				1-Aminocyclopropane-1-carboxylate oxidase homolog 1		
TraesCS1B02G117500				1-Aminocyclopropane-1-carboxylate oxidase		
TraesCS1A02G089600				1-Aminocyclopropane-1-carboxylate oxidase		
TraesCS1A02G089500				1-Aminocyclopropane-1-carboxylate oxidase		
TraesCS7D02G536900				1-Aminocyclopropane-1-carboxylate oxidase homolog 2		
TraesCS1B02G117400				1-Aminocyclopropane-1-carboxylate oxidase		
TraesCS2D02G394200				1-Aminocyclopropane-1-carboxylate synthase		
TraesCS7D02G127600				Ethylene-responsive transcription factor RAP2-9	*Arabidopsis thaliana*	[60]
TraesCS2B02G127800				Ethylene-responsive transcription factor RAP2-3		
*Carriers for transport of substrates and S-type anion channel*						
TraesCS5A02G300800				Mitochondrial dicarboxylate carrier 1	*Arabidopsis thaliana*	[61]
TraesCS5D02G307000				Mitochondrial dicarboxylate carrier 1		
TraesCS5B02G300300				Mitochondrial dicarboxylate carrier 1		
TraesCS3D02G157800				Mitochondrial arginine transporter BAC2	*Oryza sativa*	[62]
TraesCS3B02G177000				Mitochondrial arginine transporter BAC2		
TraesCS3A02G225100				S-type anion channel SLAH3	*Arabidopsis thaliana*	[63]
TraesCS3B02G254700				S-type anion channel SLAH3		
TraesCSU02G001600				S-type anion channel SLAH2		
*Allene oxide synthase (alpha-Linolenic acid metabolism)*						
TraesCS4B02G237600				Allene oxide synthase 2	*Arabidopsis thaliana*	[64]
*Nudix hydrolase*						
novel.15168				Nudix hydrolase 17	Mammalian	[65]
TraesCS6B02G326200				Nudix hydrolase 17	Mammalian	[40]
TraesCS7B02G110900				Nudix hydrolase 21		

Three blocks in heatmap represent, from left to right, wheat seedlings treated with 7.5 mM NO_3_^−^ (CK), 7.5 mM NH_4_^+^ (SA), and 7.5 mM NH_4_^+^ + 3 mM HCO_3_^−^ (AC). Numbers in color scale bar indicate log_2_ (FC) in gene expression.

**Table 2 biology-13-00101-t002:** Fermentation and Pyr metabolism enzyme activity.

Activity (nmol min^−1^ g^−1^ FW)	CK	SA	AC	Change (SA vs. CK) (%)	Change (AC vs. SA) (%)
PDC	72.11 ± 1.17 b	98.57 ± 1.74 a	79.16 ± 3.25 b	36.69 ± 2.42	−19.69 ± 3.31
ADH	26.53 ± 1.96 c	38.12 ± 1.04 a	32.96 ± 1.99 b	43.66 ± 3.94	−13.52 ± 5.22
LDH	57.67 ± 5.04 b	93.20 ± 5.94 a	62.84 ± 5.40 ab	61.90 ± 6.97	−32.35 ± 7.66
PK	1172.05 ± 14.85 a	1039.71 ± 23.17 c	1085.18 ± 14.49 b	−11.29 ± 1.98	4.37 ± 1.39
PDH	1.72 ± 0.09 b	1.53 ± 0.08 b	2.95 ± 0.17 a	−11.15 ± 4.91	93.17 ± 11.27

Results represent mean ± SD of at least three independent experiments. Different lowercase letters in each row indicate significant differences at *p* < 0.05. Wheat seedlings treated with 7.5 mM NO_3_^−^ (CK), 7.5 mM NH_4_^+^ (SA), or 7.5 mM NH_4_^+^ + 3 mM HCO_3_^−^ (AC).

**Table 3 biology-13-00101-t003:** Effects of N treatments on concentrations of metabolites from amino acid metabolism, Pyr metabolism, TCA cycle, and GABA shunt in roots.

Concentration (µg g^−1^ FW)	CK	SA	AC
Ala	635.78 ± 2.92 c	763.37 ± 7.12 a	736.99 ± 1.46 b
formate	150.83 ± 1.14 b	240.93 ± 13.99 a	246.21 ± 7.84 a
Trp	10.73 ± 0.16 c	18.39 ± 0.04 a	13.32 ± 0.05 b
Tyr	42.90 ± 0.94 c	68.93 ± 0.01 a	60.63 ± 1.94 b
Phe	58.72 ± 0.25 c	79.61 ± 0.13 a	75.96 ± 0.86 b
Pyr	98.80 ± 0.03 a	70.12 ± 0.03 c	85.67 ± 0.30 b
acetyl-CoA	47.34 ± 2.25 a	36.25 ± 2.00 c	42.54 ± 1.69 b
CA	101.02 ± 9.28 a	51.16 ± 4.74 b	56.36 ± 0.84 b
KGA	6.99 ± 0.05 a	4.59 ± 0.23 c	5.98 ± 0.19 b
succinate	2322.80 ± 51.37 a	1173.72 ± 8.72 c	1928.36 ± 90.09 b
fumarate	196.15 ± 6.03 b	80.45 ± 0.66 c	221.67 ± 6.57 a
malate	882.76 ± 39.05 a	327.22 ± 15.67 c	683.86 ± 5.87 b
OAA	11.60 ± 0.03 a	1.79 ± 0.02 c	3.20 ± 0.30 b
GABA	311.16 ± 2.14 b	444.51 ± 0.71 a	279.76 ± 0.78 c

Results represent mean ± SD of at least three independent experiments. Different lowercase letters in each row indicate significant differences at *p* < 0.05. Wheat seedlings treated with 7.5 mM NO_3_^−^ (CK), 7.5 mM NH_4_^+^ (SA), or 7.5 mM NH_4_^+^ + 3 mM HCO_3_^−^ (AC).

**Table 4 biology-13-00101-t004:** Differentially expressed genes related to cell division and cell elongation in wheat roots under different N treatments.

Gene ID	Gene Expression 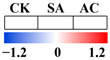			Gene Description
*Mitotic-specific cyclin and mitotic checkpoint proteins*				
TraesCS4D02G076900				G2/mitotic-specific cyclin-A3-1
TraesCS1B02G320900				G2/mitotic-specific cyclin-B1-5
TraesCS1A02G309800				G2/mitotic-specific cyclin-B1-5
TraesCS1D02G309300				G2/mitotic-specific cyclin-B1-5
TraesCS3A02G157000				G2/mitotic-specific cyclin-A1-1
TraesCS5D02G121100				G2/mitotic-specific cyclin-A2-1
TraesCS7A02G549500				G2/mitotic-specific cyclin-B2-2
TraesCS1B02G321000				G2/mitotic-specific cyclin-B1-5
TraesCS3D02G326600				G2/mitotic-specific cyclin-B1-1
TraesCS7D02G536000				G2/mitotic-specific cyclin-B2-2
TraesCS6B02G195800				Mitotic spindle checkpoint protein BUBR1
TraesCS6A02G168100				Mitotic spindle checkpoint protein BUBR1
TraesCS3B02G183400				G2/mitotic-specific cyclin-A1-1
TraesCS7B02G472800				G2/mitotic-specific cyclin-B2-2
TraesCS4B02G078300				G2/mitotic-specific cyclin-A3-1
TraesCS5B02G078200				G2/mitotic-specific cyclin-A3-2
novel.19039				G2/mitotic-specific cyclin-A2-1
TraesCS1D02G309400				G2/mitotic-specific cyclin-B1-5
TraesCS4A02G236700				G2/mitotic-specific cyclin-A3-1
TraesCS5A02G108500				G2/mitotic-specific cyclin-A2-1
TraesCS3B02G363200				G2/mitotic-specific cyclin-B1-1
TraesCS3D02G164600				G2/mitotic-specific cyclin-A1-1
TraesCS3A02G333000				G2/mitotic-specific cyclin-B1-1
TraesCS5B02G493100				Condensin-2 complex subunit D3
TraesCS5B02G114300				G2/mitotic-specific cyclin-A2-1
TraesCS3A02G523600				G2/mitotic-specific cyclin-A2-1
TraesCS2A02G404800				G2/mitotic-specific cyclin-B2-1
novel.314				G2/mitotic-specific cyclin-B1-5
TraesCS6D02G157200				Mitotic spindle checkpoint protein BUBR1
TraesCS2D02G401700				G2/mitotic-specific cyclin-B2-1
TraesCS5D02G493500				Condensin-2 complex subunit D3
TraesCS2B02G422800				G2/mitotic-specific cyclin-B2-1
TraesCS5A02G480000				Condensin-2 complex subunit D3
TraesCS5B02G401600				Mitotic checkpoint protein BUB3.3
TraesCS5A02G072000				G2/mitotic-specific cyclin-A3-2
TraesCS2A02G320000				Mitotic spindle checkpoint protein MAD2
TraesCSU02G054300				G2/mitotic-specific cyclin-A2-1
TraesCS2B02G249400				Mitotic checkpoint serine/threonine-protein kinase BUB1
TraesCS2A02G226000				Mitotic checkpoint serine/threonine-protein kinase BUB1
TraesCS5A02G396600				Mitotic checkpoint protein BUB3.3
*Cell division*				
TraesCS3D02G123900				Cell division cycle 20.1, cofactor of APC complex
TraesCS3B02G141100				Cell division cycle-associated 7-like protein
TraesCS6A02G268900				Cell division cycle-associated 7-like protein
TraesCS1D02G219200				Cell division control protein 45 homolog
TraesCS7D02G198900				Cell division control protein 6 homolog
TraesCS2A02G488400				Cell division cycle-associated 7-like protein
TraesCS6B02G296200				Cell division protein FtsZ homolog 1
TraesCS7B02G102800				Cell division cycle-associated 7-like protein
TraesCS7A02G197400				Cell division control protein 45 homolog
TraesCS3A02G121700				Cell division cycle 7-related protein kinase
TraesCS6D02G246000				Cell division cycle-associated 7-like protein
TraesCS2B02G516000				Cell division control protein 45 homolog
TraesCS5D02G149500				Cell division cycle-associated 7-like protein
TraesCS5B02G141300				Cell division control protein 6 homolog
TraesCS3D02G080400				Cell division control protein 45 homolog
TraesCS1B02G230600				Cell division cycle-associated 7-like protein
TraesCS3A02G080700				Cell division cycle 20.1, cofactor of APC complex
TraesCS4D02G123200				Cell division cycle-associated 7-like protein
TraesCS3A02G369000				Cell division cycle 20.1, cofactor of APC complex
TraesCS3B02G095100				Cell division cycle protein 27 homolog B
TraesCS4B02G129200				Cell division cycle-associated 7-like protein
TraesCS4B02G230900				Cell division cycle protein 16 homolog
TraesCS6A02G251400				Cell division control protein 45 homolog
TraesCS7D02G392900				Cell division cycle 7-related protein kinase
TraesCS5A02G142600				Cell division protein FtsZ homolog 1, chloroplastic
TraesCS3B02G400900				Cell division cycle protein 23 homolog
*Cell elongation*				
TraesCS5A02G250600				Protein activator for cell elongation 1
TraesCS7B02G118800				Protein activator for cell elongation 1
TraesCS4A02G060200				Protein activator for cell elongation 2
TraesCS7A02G559400				Cell elongation protein Dwarf1
TraesCS7B02G484200				Cell elongation protein Dwarf1
TraesCS5D02G258300				Protein activator for cell elongation 1

Three blocks in heatmap represent, from left to right, wheat seedlings treated with 7.5 mM NO_3_^−^ (CK), 7.5 mM NH_4_^+^ (SA), and 7.5 mM NH_4_^+^ + 3 mM HCO_3_^−^ (AC). Numbers in color scale bar indicate log_2_ (FC) in gene expression.

## Data Availability

The datasets used and/or analyzed during the current study are available from the corresponding author on reasonable request.

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
