# Peer review of "Bicarbonate-Dependent Detoxification by Mitigating Ammonium-Induced Hypoxic Stress in Triticum aestivum Root"

_biology, 2024, doi:10.3390/biology13020101_

Round 1

Reviewer 1 Report

Comments and Suggestions for Authors

The authors conducted an in-depth study of bicarbonate-dependent detoxification by mitigating ammo-2 nium-induced hypoxic stress in Triticum aestivum root. The authors, using transcriptomic analysis, were able to establish that  SA treatment significantly upregulated the expressions of genes encoding fermentation enzymes 17 (pyruvate decarboxylase (PDC), alcohol dehydrogenase (ADH), and lactate dehydrogenase (LDH)), 18 oxygen consumption enzymes (respiratory burst oxidase homologs, dioxygenases, and alternative 19 oxidases), downregulated the expressions of genes encoding oxygen transporters such as PIP-type 20 aquaporins and non-symbiotic hemoglobins and that involved in energy metabolism including tri-21 carboxylic acid (TCA) cycle enzymes, and ATP synthases but upregulated the glycolytic enzymes 22 in the roots, and downregulated the expressions of genes involved in the cell cycle and elongation 23 compared with the CK. The research was carried out at a high scientific level using modern methods of molecular genetic analysis.

However, there are a number of issues that need clarification. For identify differently expressed gene (DEGs) and differential transcript usage (DU), you undoubtedly used one of the pairwise comparisons methods. And then, based on the results of the analysis, a volcano plot was built, apparently using the R programming environment. Since there are no scripts for your calculations in the supplemental material and there is no relevant information in the materials and methods, it is difficult to conclude which calculation methods were used and which packages were used. Maybe VOOM function from Limma package or something else? Please describe this in the materials and methods.

As well as:

1. Please read the text of the article carefully: references are given only to literary sources No. 1-84.

2. Please match the number of literary sources in the list (102) with the number of sources cited in the text of the article.

3. Please note that the name of table 2 is written in different fonts.

4. «Figure 10. RNA-Seq data accuracy verification.» The caption to the picture does not match the picture itself.

Comments on the Quality of English Language

The quality of English language is sufficient

Reviewer 2 Report

Comments and Suggestions for Authors

Comments for Authors

I appreciate the authors Liu et al. conducted a comprehensive study on the impact of ammonium (NH4+) toxicity in wheat plants and the potential alleviation of this toxicity through bicarbonate (HCO3-). However, I have some technical questions and general comments before the manuscript endorse for publication in Biology.

The abstract is generally well-written, but in some instances, it could benefit from greater clarity and precision. For example, the term "oxygen consumption enzymes" might be more informative if specific enzymes were mentioned, directly. Moreover, in the physiological assay, instead of a general discussion of activities like PDC, ADH, and LDH, as well as root ethanol concentration and lactate efflux, presenting the changes as a percentage comparison would enhance clarity. This approach will make it more understandable by indicating the degree of increase or decrease in a more quantitative manner.

In materials and methods the authors mentioned that in our preliminary experiment, eight-day-old seedlings grew best in 394 half-strength Hoagland nutrient solution, among other details. It would be more suitable if the authors provide that data in a supplementary file for reference. Additionally, I have a question regarding the choice of a control group (CK) with nitrate (NO3-). It would be helpful to elaborate on why this control was chosen and how it compares to the treatments. What is the rationale for choosing this CK in interpreting the effects of NH4+ and HCO3-?

Generally, the discussion mentions significant changes in gene expression, but it would be beneficial to highlight specific genes or pathways affected. This could enhance the understanding of the molecular mechanisms involved. Moreover, by discussing the alleviating effects of bicarbonate, it will be more logical to discuss the broader implications of these findings for the productivity of wheat plants.

The authors discussion is overly descriptive and repetitive of the findings throughout. It is suggested to enhance the scientific rigor by minimizing unnecessary repetition and reducing the number of in-text citations. 

Overall, the discussion did not illustrate the unexpected findings or limitations to enhance the research vitality and help readers interpret the results more accurately. Moreover, no information was provided on specific targets identified in the study that could be further explored for crop improvement strategies.

Round 2

Reviewer 2 Report

Comments and Suggestions for Authors

No additional comments the manuscript is endorsed for publication.

Comments on the Quality of English Language

Minor editing